# Nanotechnology in Residential Building Materials for Better Fire Protection and Life Safety Outcomes

Charmaine Mullins-Jaime [1,*] and Todd D. Smith [2]

1   Department of Built Environment, College of Technology, Indiana State University,
    Terre Haute, IN 47809, USA
2   Department of Applied Health Science, School of Public Health, Indiana University—Bloomington,
    Bloomington, IN 47405, USA
*   Correspondence: charmaine.mullins-jaime@indstate.edu

**Abstract:** Residential fires are the main source of fire deaths and injuries both in the United States and globally. As such, better fire-resistant building materials are needed to bolster fire protection and to enhance life safety. This is during a time when fewer materials are being used to construct homes. Nanotechnology may be a solution if it can overcome its current barriers to widespread adoption in residential construction, namely economy, sustainability, and safety. This research effort includes a critical examination of the literature from a safety perspective to address fire deaths and prevent personal injuries and illnesses by targeting fortification of residential construction building materials via the use of nanotechnology. The paper reviews nanotechnology for building materials by material type, known toxicity of various nanomaterials used in construction, and a discussion on a way forward through assessing materials by their ability to satisfy the requirements of sustainability, economy, and safety- both as a material designed to reduce fire injury and death and from a toxicological hazard perspective.

**Keywords:** nanotechnology; fire-resistant; construction; building materials





## 1. Introduction

With global trends of population growth, high cost of resources and a growing necessity for sustainably sourced building materials, modern buildings are often made with fewer resources, designed to be lighter, stronger, and more durable [1]. However, without additional treatments, this can also mean they are less fire-resistant [2], particularly, if those materials are not fortified to meet fire resistance requirements such as those set out by the National Fire Protection Agency (NFPA) or in local, national, or international building codes. Fire-resistant materials are especially important in residential construction from a life safety perspective since residential fires are the leading source of fire deaths in the United States [3,4] and around the world [5]. The U.S. Fire Administration's US Fire Statistics from the years 2010–2019 indicate while fire incidents are down 3.2% from 2010, the incidence of fire deaths has increased by 24. 1%. Residential fires were the second most frequent type of fire, next to outdoor fires, and the leading property type for fire deaths (72.2%), fire injuries (76.4%), and fire dollar loss (46.4%) in 2019 in the United States [4]. This trend is echoed globally where, in 2020, residential fires were also the leading property type for fire deaths (82.7%) and injuries (60.6%) [5].

While fire prevention is a primary mechanism of fire death prevention, prompt and effective alarm, alarm response and safe egress are critical components of life safety [6]. Fire resistance in building materials is a growing concern as it is crucial for fire response and egress. Variables such as fire resistance, material durability, structural strength, smoke release, and toxicological hazards in building materials are critical in protecting inhabitants and allowing them a better chance of escape and survival in the event of a fire. However, many manufacturers design products to make the most profits, including using lighter,

thinner materials, regardless of their flammability [1]. In addition to use of lighter thinner, and less fire-resistant materials, certain flammable materials are used in building construction such as outside heat insulation panels and phase change materials used for cooling buildings, which is leading many researchers to find innovative solutions to this growing problem. Examples of such progress is use of triazine-based compounds to produce intrinsically flame retardant phase change materials [7] and use of biobased materials to impart fire retardant properties [8].

With emerging global trends, modern building materials in residential construction must meet a triad of requisites that fall under the umbrella of "sustainability" to be widely adopted as practical options; they will have to be economical, sustainable to the environment, and safe. Use of nanotechnology in building materials may be an attractive solution to meet these demands. Nanotechnology is the manipulation of materials on the molecular and atomic scale. Nanomaterials have one or more dimensions in the 1–100 nm range [9]. Their nanoscopic size allows for coatings and material composites that are stronger, less porous, and lightweight.

Nanomaterials can be used in building materials such as coatings, steel, clay composites, concrete, insulation, and windows. While nanomaterial use in building materials is not widespread [10,11], demand is expected to grow due to global economic and population growth driving demand for building materials [1]. While the application of various nanotechnology in building materials under fire conditions has been reviewed previously in the literature, this paper takes a public safety perspective to address fire deaths and prevent personal injuries and illnesses. The following is a narrative review of nanomaterials in building construction, their known toxicity, and a discussion of a way forward for safe and pragmatic adoption of nanomaterials in residential construction for better life safety outcomes and fire protection.

## 2. Methods

Databases Google Scholar as a primary search engine and EBSCOhost, JSTOR and the Indiana State University Library as secondary search engines were used with no restrictions on country or publication date. Search terms included the following: nanotechnology, nanomaterials AND construction, construction materials, building materials; nanotechnology, nanomaterials AND fire protection. Additional search terms were used for specific construction nanomaterial and fire protection as follows: nanomaterial concrete, nanomaterial steel, nanomaterial wood, nanomaterial insulation, nanomaterial coatings AND fire protection.

Health hazards of nanomaterial in construction were found with the following search terms: nanotechnology building materials, nanomaterial construction AND toxicological hazards, health effects, health hazards.

Additional search terms were used for toxicological hazards associated with nanomaterials as follows: nanomaterial carbon nanotubes, nanomaterial graphene, nanomaterial carbon black, nanomaterial silver, nanomaterial titanium dioxide, nanomaterial zinc oxide, nanomaterial copper oxide, nanomaterial zinc oxide, nanomaterial iron oxide, nanomaterial copper oxide, nanomaterial silica AND toxicological hazards. Nanomaterial toxicological exposure assessment methods were found with the following search terms: toxicological hazard assessment methods; risk assessment methods, exposure assessment methods AND nanotechnology; nanomaterials. Relevant articles were also found through backward search by scanning the references of found articles.

Articles were included if they were written in the English language and if they were relevant to the topics reviewed in this paper namely: nanomaterial use in building construction, nanomaterial use for fire protection in building construction, toxicological hazards of nanomaterial used in building construction, and hazard assessment of nanomaterials. Additional literature on toxicological hazards associated with specific nanomaterials were included if those materials were also used in construction.

### 3. Nanomaterials in Building Construction

*3.1. Concrete*

Silica, titanium dioxide, iron oxide, and carbon nanotubes are common nanomaterials added to concrete to increase density, strength and in turn, fire resistance [11–13]. Silica particles are added to concrete to fill the voids between cement grains making the concrete denser and less porous. This results in increased mechanical strength [13]. There are two types of nanomaterials used in concrete, silica fume and nano-silica—also known as "fumed silica" [10]. Nanosized titanium dioxide can be used in concrete for self-cleaning and removal of toxins [12]. Use of iron oxide nanoparticles can improve compressive strength and abrasion resistance [13]. Carbon nanotubes (CNTs) in concrete have been identified as having the potential to produce concrete that is strong, electrically conductive, and self-healing [12]. A novel and promising development in nanotechnology in concrete their ability to impart self-sensing properties using carbon nanotubes or carbon black nanoparticles that have implications for better assessing a building's structural safety during or after a fire [14–16], as is the use of nanomaterials that aid in post-fire curing of concrete that improve capability to recover strength post-fire event [17,18].

*3.2. Windows and Glass*

Nanomaterial film added to glass can provide insulation, self-cleaning and fire-resistant properties [11,12]. Fire safety glass is another type of nanomodified material, using either silica fume or nano-silica, that can provide high levels of fire protection by creating an intumescent layer between two plates of glass [11,12]. During a fire, the intumescent layer expands and turns opaque, providing a high level of integrity and insulation. This type of glass has been available for over 30 years, but mainly where a high level of thermal insulation is required such as escape routes [11].

*3.3. Insulation Materials*

Silica-based aerogels can be used in insulation blankets, translucent windows, or vacuum-insulated panels and are highly effective thermal insulants [10]. Production of these materials is expensive and not widely used [11,12]. Poor mechanical strength [19] is another barrier to their widespread adoption. However, costs could be reduced with improved production methods and scale as demand for energy-efficient construction materials increases.

While use of nanotechnology as insulants are dependent on their ability to impart low thermal conductivity of the building material, certain uses of nanomaterial in construction can actually increase thermal conductivity which may be problematic for fire protection unless the material composite also has fire resistant properties.

*3.4. Steel*

Nanomaterial can be used in steel as a nano-coating or incorporated into its fabrication. Nanocomposite polymers and coatings applied to the steel structures can reduce the heat released and improve fire retardancy [20]. By refining materials down to the nanoscale and driving out impurities such as carbides, the steel becomes stronger, more resistant to corrosion, and possesses a tensile strength 100 times that of steel [20]. Manufacturers claim using nanomaterial in steel is as effective as stainless steel but cheaper and more effective than more traditional methods of protecting steel against corrosion such as epoxy coatings [10].

When steel structures are exposed to high temperatures, their strength and rigidity are at risk [2]. To protect occupants and reduce loss, structural steelwork usually requires fire protective materials such as cement-based sprays, boards, batt materials, and intumescent coatings, however, steel coatings tend to lose durability [21]. Nanomaterials on structural steel can provide excellent fire resistance [21,22]. In a fire, a structure may collapse if the critical temperature is reached. During a fire, improved steel integrity would allow prolonged time before collapse, giving occupants more time to escape, and can ensure the

integrity of the structure during and after a fire, making it safer for rescue workers and leaving less property damage [1].

### 3.5. Wood

Wood is one of the most widely used building materials and the most common residential construction material used in North America. Timber as a building material has good mechanical properties and can be an indefinite renewable resource if forest resources are properly managed, and can serve to store atmospheric carbon, potentially for long periods of time, if the life of timber products is extended [23]. However, wood is a less resilient and fire-resistant building material than non-renewable materials such as concrete and steel.

Untreated wood is highly combustible. Building codes limit wood use mainly to residential construction due to its low fire resistance, unless treated with fire-resistant coatings. Fire resistance can be improved with traditional chemical fire retardants. However, traditional fire-resistant coatings can produce toxic gas [24]. The chemicals are associated with environmental and health risks and are less effective than nanomaterial in wood composite or by use of nano-coatings made with nanoclay and oxides $SiO_2$ and $TiO_2$ [25]. Further, most fire-retardant coatings have poor resistance to external factors and weathering [26], thus better technologies are needed. Vakhitova's review of fire-resistant coating for wood found the APP/PER/MA intumescent system is the most reliable and economically viable and suggests it could be enhanced with synthesis of nanoclay, nanostructured carbon, or amorphous silicon dioxide [24].

Researchers are working on nanotechnology wood coatings that can provide fire-resistant properties, while posing benign health risks and have created nano-coatings that have imparted fire-resistant properties on wood [27–31]. One interesting development in wood construction fire protection is hydrothermal synthesis of nanooctahedra $MnFe_2O_4$ onto the wood surface that creates a fire-resistant and electromagnetic wave-absorbing coating [32]. This might be a viable option from a safety perspective since manganese ferrite is already used in nanoparticle-based drugs, it is soluble, so it will not remain in the respiratory tract if inhaled, and there is evidence that the body can easily process and remove it without organ damage [33].

Another interesting development is the use of an environmentally benign polyelectrolyte complex that, when coated on wood, provides fire resistance, self-extinguishing behavior, increased time to ignition, and reduced peak heat release rate [34]. This coating also increased the strength of the wood. These potential new construction material technologies could provide vast societal benefits by providing better fire resistance for wood construction, which can prevent injuries and fatalities, and minimize economic loss. In sum, nanomaterials in wood construction can prove a superior and sustainable source of fire-resistant building materials but more research is needed to better understand lifecycle and toxicological effects.

### 3.6. Coatings and Composites

Nanocoatings can be applied to various building materials. When applied to paint and drywall, they form an intumescent layer. When exposed to heat, the intumescent layer creates a char. This acts as a fire retardant because char is a poor conductor, offering better fire protection for the material behind it.

Nano-sized Boron Nitride (BN) and micron-sized BN used as fillers in fire-resistive coatings were found to be effective in enhancing fire-resistive coating's thermal stability, especially under high temperature [35]. In a comparison between organic and inorganic intumescent coatings, Wang et al. [22] found organic intumescent coatings have a good expanding effect and char structure, but they generate solvent toxic gas and smoke in a fire. Inorganic intumescent coatings such as salt silicate coatings do not create organic solvent in application and have little toxic gas emissions and smoke when heated. However, inorganic intumescent coatings are vulnerable to moisture and give fire protection only

at a low temperature [19]. There is still much research needed to understand the product lifecycle and toxicological effects of nanocoatings [36]. However, the fire safety implications of nano-coating materials are enormous.

Nanocomposites such as nanoclay can be used to fortify building materials and aid in fire resitance. Nanoclays can be used as coatings and as composites in building material. Benefits of nanoclays are increased density and strength of building materials [37]. Nanoclays can be naturally occurring or synthetic, formed with layers of silicate-based materials, and can be used in polymers to improve functionality in various ways, but there are currently no products available for use in construction [10]. Nanoclay brick has vast potential benefits in fire protection if introduced into the construction material market [12].

## 4. Toxicological Hazards Associated with Nanomaterials Used in Construction

Hazards of nanoparticles are dependent on size, shape, aggregation, agglomeration, solubility, electrical charge, and toxicity [10]. Nanoparticles are primarily absorbed through the respiratory system [38]. Particles smaller than 10 nm are retained in the respiratory tract whereas particles with dimensions 10–100 nm are deposited in the bronchioles and alveoli [39]. Nanomaterials can also be absorbed through skin contact and ingestion [40].

As particles get smaller, their surface area to mass ratio increases. This is associated with an increase in their reactivity. The shape of the particle plays a critical role. Fiber-shaped particles such as those found in carbon nanotubes (CNTs) are a particular concern [10]. Because of their microscopic size, they can travel to the deepest regions of the lungs and, because of their long shape, like asbestos, they can become trapped in the lower respiratory tract [11]. Airborne nanoparticles can enter the bloodstream 15 min after inhalation because nanoparticles are able to break the barrier between the lungs and the bloodstream, increasing the risk of a heart attack or stroke [39].

How particles stick together also influences the extent the particle can penetrate deep into the lungs and their solubility affects whether they remain in the lungs. Insoluble nanoparticles can penetrate the lower respiratory tract making it difficult for the lungs to clear. This can cause inflammation of the lung as well as inflammation in other body parts or can be carried through the circulatory system to other organs. Nanomaterial with water solubility higher than 100 mg/L, are considered to be sufficiently soluble and can be assessed by traditional chemical toxicity since they are no longer nanomaterials after they have been absorbed by the body [12,41].

In many cases, toxicity can be assessed based on traditional toxicology by evaluating dose–response of the parent material in non-nano form [12,42–44]. However, some nanoparticles have been found to be more harmful than their larger scale counterparts. For example, nanoscale titanium dioxide particles have higher mass-based potency than larger particles and should be treated as a potential occupational carcinogen [40]. The surface area and the number of particles in the air are a better measure of exposure than the mass concentration [36]. Further, thorough life cycle analysis (LCA) to assess the environmental impacts of a nanomaterial, from a cradle to grave perspective, are needed to understand when and how the material may be harmful to people and the environment. While these assessments on certain nanomaterials used in construction are available in the literature [45–47], studies applying LCA to nanotechnology are limited [47]. Considering LCA is a critical component in anticipating how individuals may be exposed to toxicological hazards associated with nanomaterial.

The following is a summary of known toxicity of nanomaterial used in building construction.

### 4.1. Carbon Nanotubes (CNTs)

Carbon nanotubes (CNTs) are hollow structures constructed from layers of rolled graphene [10]. They can be several microns or longer. Their tiny diameter and longer fiber-like length make them potentially toxic if they have similar shape and structure to asbestos: long, thin, and insoluble. Hard insoluble CNTs longer than 5 μm are more toxic, can embed into the lung tissue, and are difficult to expel than shorter or tangled CNTs [8].

However, some studies have identified that shorter fibers (i.e., less than 2 μm) can still have adverse effects, and there is also a lack of agreement on the diameter of CNTs which are the most problematic [10]. The adverse health effects are inflammation, respiratory distress, clotting, fibrosis, vascular damage, and other cardiovascular effects [10,13,48,49].

Carbon nanoparticles with dimensions below 100 nm were found to remain in the respiratory system for long periods. One study showed only 25% of the deposited nanoparticles were removed from the respiratory system within 24 h and retention time of 75% of inhaled particles exceeded 48 h [50]. This long retention time promotes their penetration into the epithelial cells of the respiratory tract, bloodstream, or lymphatic system [39].

### 4.2. Graphene

Graphene carbon nanomaterial can be used in coatings, electrodes, transistors, and concrete. Research into the health effects is at an early stage [12]. There are conflicting opinions on toxicity of nanoparticle graphene [51]. There is literature indicating graphene materials are biocompatible [52–54], while other studies have reported adverse effects and cytotoxicity [55–57].

### 4.3. Carbon Black

Carbon black (CB) is found in rubber manufacturing and can be used in construction in coatings to shield against radio waves or to provide electrical conductivity. Carbon black is classed as a possible carcinogen and associated with irreversible respiratory illnesses [10]. Biological effects of carbon black nanoparticles vary in different target cells and are determined by a combination of surface area and composition of surface-bound polycyclic aromatic hydrocarbons [58].

### 4.4. Silver

Silver is known for its antimicrobial properties. Silver nanoparticles can accumulate in various organs including liver, kidney, and brain and can lead to cell death [39]. They can also damage the blood–brain barrier, cause inflammation and damage DNA [39,59,60].

### 4.5. Titanium Dioxide

Ultra-fine particles of titanium dioxide are a suspected occupational carcinogen. In their 2011 Current Intelligence Bulletin 63, the United States National Institute for Occupational Safety and Health (NIOSH) recommended a maximum workplace exposure level for nanoparticle titanium oxide as one-eighth of the exposure level for non-nano titanium dioxide [61]. Titanium dioxide is considered to be neurotoxic and can also accumulate in the nervous system of fetuses [39]. It is also associated with inflammation in the lungs, changes in DNA, metabolic changes, and cell death [13].

### 4.6. Zinc Oxide, Iron Oxide, and Copper Oxide

Zinc oxide can be used in construction and window coatings. There have been few studies of nanomaterial zinc oxide toxicity however there is some evidence it is linked with inflammation, cell proliferation, and cytotoxicity [13,62,63]. Iron oxide and copper oxide nanoparticles can damage DNA [13]. Copper oxide nanoparticles are neurotoxic and tend to be more neurotoxic than other metals [39].

### 4.7. Silicon Dioxide (Silica)

Unlike crystalline silica, which is associated with severe adverse health effects in the construction industry, nanosilica is usually amorphous [12]. Nano silica aerogels appear to commonly use amorphous silica [64,65]. Amorphous silica nanoparticles are considered to be relatively low risk, however, there is insufficient evidence to declare nanosilica as 'safe' [12,66], as there is some evidence of cytotoxicity [13,67].

## 5. A Way Forward in Assessing Viability of Fire-Resistant Nanotechnology in Residential Construction

Use of nanotechnology in building materials can be highly effective in providing fire-resistant and structure-strengthening properties. However, their current barriers to widespread adoption are cost and the need for assurances of their safety since there is limited information available on the toxicological effects of nanomaterial exposure and limited provisions for their control [1,20,68–70]. These same barriers are also likely drivers of mainstream adoption of nanomaterials in residential construction as discussed below.

Because the majority of fire deaths occur in residential properties both in the United States and around the world and to address life safety in fire protection, better fire-resistant materials in residential construction are needed. In North America and many other countries around the world, better protected wooden construction will have maximum impact on fire death and injury prevention as this is a primary material used in residential construction [23,34]. Use of nanotechnology in building materials has significant potential to create safer and more resilient residential structures, however, to overcome barriers to widespread adoption, certain criteria must be met.

To assess the viability of a particular nanomaterial product to be adopted for residential construction, it is important to understand its strengths and limitations compared with current fire-resistant coatings available on the market. They must also be assessed by their ability to satisfy the requirements of sustainability and its three pillars as adopted by the United Nations and United States Environmental Protection Agency: economy, environment, and social/people—primarily from a social construct are assurances of safety, both as a material designed to reduce fire injury and death and from a toxicological hazard perspective. These variables are deeply interdependent on one another, as discussed below, for the viability of a particular nanotechnology to emerge as an ideal solution to improve life safety outcomes in fire protection for residential construction.

### 5.1. Sustainability: Environmental and Economic Considerations for Nanotechnology in Building Materials

Beyond altruistic intentions of using a renewable material or one that has low environmental impact, a more likely practical driver of renewable and less impactful product demand will be as costs of non-renewable resources increase and material availability of non-renewables decrease, builders will gravitate to more pragmatic options to manage costs and ensure a steady supply of building materials. Thus, considerations for economy are interconnected in ensuring a robust and cost-effective supply chain. Simplicity and scalability of production are other important economic factors in viability of bringing a nanotechnology building material to market. However, cost can be driven down by process improvements and in sourcing lower-cost options in their synthesis or manufacture. Lowering cost through economies of scale will be heavily dependent on mainstream demand which is partially contingent on assurances the product is safe [1,69,71]. No one wants to re-create the tragic mistakes of the past such as the use of asbestos or lead in construction.

### 5.2. Sustainability: Safety and Environmental Health, Both Product Safety and Minimization of Toxicological Hazards and Environmental Impacts of Nanotechnology in Building Materials

To justify widespread adoption of a particular construction material, it should show superior fire resistance, structural strength, durability, release less hazardous smoke, and finally, present fewer toxicological hazards than conventional coatings and materials.

### 5.2.1. Assessing Building Material for Fire Safety

ASTM E119-20 Standard Test Methods for Fire Tests of Building Construction and Materials [72] and Underwriters Laboratory UL 263 Fire Tests of Building Construction and Materials [73] are examples of guidance materials that can be used to determine material fire resistance. ASTM D4761-19 Standard Test Methods for Mechanical Properties of Lumber and Wood-Based Structural Materials [74] is an example of guidance that can be used to evaluate structural strength of material. An example of guidance that could be used to

assess material durability is Underwriters Laboratory UL 2431 Standard for Safety for Durability of Fire Resistive Coatings and Materials [75] ASTM E1354-22a Standard Test Method for Heat and Visible Smoke Release Rates for Materials and Products Using an Oxygen Consumption Calorimeter [76] is an example of a standard that can be used to evaluate smoke release.

### 5.2.2. Assessing Building Material for Environmental and Toxicological Hazards

There has been a significant amount of research on the development of nanotechnology and concern for health [77]. Knowledge of nanomaterial risks is based on laboratory research vs. reported cases of illness [10], it is uncertain if this is indicative of nanomaterial safety or long latency periods like those of asbestos exposure. Furthermore, there are multiple assessment methods and different material compositions making comparison of results across studies difficult [12,43]. Other than some specific recommended exposure limits for nanoform $TiO_2$, CNT and nanofibers, information on exposure thresholds is limited [78].

Life cycle analysis of all aspects of their potential to create harm from inception to final disposal are an ideal framework for their assessment. However, while there are various provisions such as those under EU's REACH, US Toxic Substance Control Act, and from regulators such as OSHA, there are limited regulatory framework guidance and enforcement to ensure the responsible development of nanotechnology [1,17,66,67]. Without clear direction and a regulatory requirement to do so, there is little incentive for manufacturers to put efforts into anticipation, assessment, and control of nanoparticle hazards to protect both the public and the employees who manufacture the products. However, recent amendments to EU's REACH attempt to address this problem by requiring material hazard assessments, but this is not applicable to materials made, used, or sold outside the European Union. Thus, manufacturers must ensure toxicological hazards are assessed thoroughly for each product/technology and product purchasers/builders must share in this onus by ensuring these hazard assessments are completed prior to mainstream adoption for residential construction.

Some researchers note risk of nanomaterials can be assessed and managed using existing toxicity models, starting with whether the non-nanoform substance is known to be hazardous [12,42–44]. However, traditional industrial hygiene approaches use a mass-based method which may not be suitable if the material poses additional health risks due to aspects such as smaller size, large surface-to-volume ratios, and material differences [12,43], as is the case for nanosized $TiO_2$ and carbon nanotubes. Thus, assessments using surface area and number of respirable particles is recommended [40]. Furthermore, differentiating solubility, shape, size, and agglomeration will also be important in assessing the toxicological hazards of a nanomaterial. Taking an industrial hygiene approach, exposure levels could be assessed quantitatively and qualitatively under certain test conditions such as during a fire, when material is cut, sanded, etc., and compared with known biological outcomes associated with the found exposure level. Industrial hygiene guidance such as the American Conference of Governmental Industrial Hygienists Threshold Limit Values TLV® and Biological Exposure Indices BEI® [79] is an example of guidance available on exposure limits and toxicological effects.

## 6. Conclusions

Residential fire deaths and injuries are a growing safety and public health concern. Better fire-resistant building materials are needed in residential construction to bolster fire protection and to improve life safety outcomes. This review examined literature, from a safety perspective, to address these concerns by examining the application of nanotechnology to building materials to improve safety. The review specifically addressed and provided guidance on nanotechnology applications related to construction materials including concrete, windows and glass, insulation, steel, wood and coatings and composites. Concurrently, the review noted potential and existing hazards that need to be addressed to satisfy the requirements of sustainability including economy, environment, and safety.

Potential and existing hazards were presented for carbon nanotubes, graphene, carbon black, silver, titanium dioxide, silicon dioxide and zinc oxide, iron oxide, and copper oxide. As noted, nanotechnology has enormous potential to improve fire safety in residential construction and prevent unnecessary injuries and deaths if barriers to widespread adoption are overcome. Guidance on a way forward was offered. Thorough assessments of viability, driven by economy and functionality, are deeply connected to their safety and thus safety must be at the forefront of emerging nanotechnologies in residential construction.

Regardless of the contributions of this review, additional research explorations and research related to nanotechnology applications to construction and nanomaterial use in construction are needed. As this research provided hazard information of nanomaterials generally related to public health and safety, more research is needed from a building construction safety and risk management perspective. Particularly, this work should aim to address hazards during the construction phase and the remainder of the building life cycle. Further, following guidance from Gorsuch and Link [80] additional policy related research is needed. Given where we are currently, more information is needed regarding the adoption of nanotechnologies and its long-term consequences. As such, longitudinal studies are warranted, especially as researchers seek to understand the impact of nanotechnologies on ecosystems once adopted.

**Author Contributions:** Conceptualization, C.M.-J.; investigation, C.M.-J.; writing—original draft preparation, C.M.-J.; writing—review and editing, C.M.-J. and T.D.S.; supervision, C.M.-J. and T.D.S.; project administration, C.M.-J. and T.D.S. All authors have read and agreed to the published version of the manuscript.

**Funding:** This research received no external funding.

**Institutional Review Board Statement:** Not applicable.

**Informed Consent Statement:** Not applicable.

**Data Availability Statement:** Not applicable.

**Acknowledgments:** The authors would like to thank Jan K. Wachter and Christopher A. Janicak for their input on preliminary versions of this work. Furthermore, the authors would like to thank the reviewers for their helpful comments and advice on expanding future research in this area.

**Conflicts of Interest:** The authors declare no conflict of interest.

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
