# Peer review of "Nanotechnology in Residential Building Materials for Better Fire Protection and Life Safety Outcomes"

_fire, doi:10.3390/fire5060174_

Round 1

Reviewer 1 Report

This paper studies the nanotechnology in residential building materials. My comments are the follows:

1.      The Abstract should be updated to highlight the motivation and novelty of this paper.

2.      Page 2, line 75-76, what is the “silica fume or fumed silica”?

3.      In Section 2.3, Due to its ultra-low thermal conductivity, silica aerogel can be used as a building insulation material. The summary on the application of silica aerogel to thermal insulation is incomplete. The reason that restricts the wide application of silica aerogel is not only the high production cost, but also the low mechanical strength. More references are needed to be added here. The following references may be cited.

[1] Effective structure of aerogels and decomposed contributions of its thermal conductivity.

[2] Characteristics of nanoporous silica aerogel under high temperature from 950°C to 1200°C.

4.      In section 3.7, please distinguish between the definitions of silica and silica aerogel. Please add a discussion on whether silica aerogel nanoparticles pose a respirable risk.

5.      Please check the format of all references.

6.      The language requires professional proofreading, as well as the article format.

Author Response

Dear reviewer 1,

Thank you for your feedback. We have revised based on your comments and believe the manuscript has been improved. 

Reviewer 2 Report

This paper revised different contributions of nanotechnology to improve the fire safety of building materials. However, the authors failed to meet important aspects of review papers. Originality of this review paper is not clear (previous works have already revised nanomodified building materials under fire situation). Some important topics were not revised in this paper. Future research directions should be indicated based on the revised state-of-the-art. The following corrections are required before publication of this manuscript:

1. Section 1: Fire statistics presented in the first paragraph could be complemented with statistical data recently published (2021-2022) in the literature.

2. Section 1: At the end of the introduction section, the authors should clarify the novelty (original contributions) of this paper, considering all review papers that already revised the application of nanotechnology in building components under fire conditions. For example, the following review papers: Sikora et al. 2018 (DOI: 10.3390/nano8070465), Rabajczyk et al. 2021 (DOI: 10.3390/ma14247849), Laím et al. 2021 (DOI: 10.1016/j.jobe.2020.102008), etc.

3. A new section should be added to describe the methodology used in this review paper. In this section, the authors should mention the review methodology used to select the papers of the reference list (e.g., review strategies, keywords, databases, etc). The authors should also mention the criteria (e.g., journal, publication year, number of citations, main topics, etc) used in order to define whether a material is relevant to this review paper.

4. Section 2: Section 2.1 revised the effects of nanomaterials on mechanical, microstructural, self-healing and self-cleaning properties of concrete. It should be complemented with a review of the effects of nanomaterials on the self-sensing properties of smart building materials subjected to fire condition (e.g., Dong et al. 2020 - 10.1016/j.cemconcomp.2020.103675; Nalon et al. 2021 - DOI: 10.1016/j.cemconcomp.2021.104104; Jang et al. 2022 - 10.1016/j.jobe.2021.103816), which is a novel and promising application of nanotechnology on structural fire safety.

5. Section 2: The contributions of nanomaterials to the post-fire curing of concrete could be revised in Section 2.1. This is another important aspect related to the contributions of nanotechnology to the fire safety engineering.

6. Section 2: What is the relationship between concrete and the weldability property cited in line 69? It seems that this property is not associated with the content of this subsection.

7. Section 2: The content of line 72 is not clear. Is glass a nanomaterial? The terms "Fire safety glass is another type of nanomaterial" could be replaced by "Fire safety glass is another type of nanomodified material".

8. Section 2: Although some insulation materials containing nanomaterials were mentioned in Section 2.3, the authors did not revise the increases in thermal conductivity provided by many types of nanomaterials incorporated into building components. For example, the addition of carbon black nanoparticles, graphene or carbon nanotubes to building elements causes increases in their thermal conductivity. The advantages and drawbacks associated with these increases should be revised in this paper.

9. Section 2: All subsections of Section 2 lack quantitative data regarding the benefits provided by the nanomaterials to construction elements. According to the findings of the revised literature, these subsections must indicate percentage variations in different properties (e.g., fire resistance, combustibility, thermal insulation, self-extinguishing, etc) caused by the inclusion of nanomaterials.

10. Section 3: This section must be significantly improved because it did not revise hazards associated with nanomaterials from the building construction perspective. It only presented a list of health issues associated with different types of nanomaterials. This list has been already revised in many review papers that are not related to the civil engineering or fire safety fields. Health AND ecological hazards associated with nanomaterials must be revised in terms of potential damages during the different steps of the construction phase and during the useful life of building environments.

11. Section 3: This section could revise the life cycle assessments (LCAs) reported in previous literature that dealt with nanomaterials applied to construction materials. For example, the following papers discussed LCAs of nanomodified building materials: Jayapalan et al. 2013 (DOI - 10.1016/j.cemconcomp.2012.11.002); Sackey et al. 2019 (DOI - 10.3390/app9071315); Hischier and Walser 2012 (DOI - 10.1016/j.scitotenv.2012.03.001).

12. Section 4: The rational structure mentioned in this section (requirements of sustainability, economy, and safety) is not appropriate. The framework presented in Figure 1 and the title of the subsections of Section 4 should be reformulated considering that the definition of sustainability is based on three main pillars (economic, environmental, and social pillars).

13. Section 4: Figure 1 is very simple and should be removed. All information provided in this image was clearly described in the text of Section 4.

14. Section 5: A brief conclusion was provided in a single paragraph. The authors should elaborate an effective conclusion section that lists the main contributions (originality) provided by the present review paper.

15. Section 5: Review papers are usually elaborated to provide future directions in the scientific field. This manuscript should be complemented with a list of topics that were not investigated in the revised literature. Recommendations for future research must be clearly indicated at the end of the paper.

Author Response

Dear reviewer 2,

Thank you for your feedback. We have revised based on your comments and believe the manuscript has been improved. Attached are responses to reviewers comments. 

Reviewer 3 Report

Reviewer comments on Manuscript ID 1964932-

 In the work by Charmain et al, they reviewed how the nanomaterials affect the buildings in terms of fire protection and life safety. The overall structure looks good, covering different topics in fire protection in buildings, such as Nanomaterials in Building Construction, Hazards Associated with Nanomaterials, and A Way Forward in Assessing Viability of Fire-resistant Nanotechnology, and others. The manuscript is well written and provide the general knowledge and progress in building fire protection. However, the following concerns should be addressed so that it could be accepted to be published.

1.     In the main text, it contains so many sections and sub-sections, the author is recommended to add an “Outline” section in front of “Introduction” section. This will help the readers easily capture the overall structure and key information.

2.     In the part of “Nanomaterials in Building Construction”, the authors listed “Concrete, Windows and Glass, Insulation Materials, Steel Construction, Nanoclays, Coatings, Paint and Drywall, Wood as sub-sections. Some of the sub-sections are grouped by building components, such as wool, concrete, windows etc. While some are based on functions or types of materials, like nanoclay, paint, insulation materials etc. The sub-sections need to be grouped based on either building components, or material type or functions.

3.     Since this review is about fire protection, the authors are recommended to add one more paragraph about the flammable materials in buildings in front of the paragraph of “With emerging global trends, modern building materials in residential construction…” at page 2.

In this paragraph, the author can introduce some of the materials investigated or used in buildings, while the materials are flammable. Such as 1) It is known that a lot of materials, such as the heat insulation panels, which are used on the outside of buildings, 2) phase change materials, which are also studied or used for building cooling etc. However, some of organic phase change materials are flammable. Some good articles on how to reduce the flammability are recommended to be cited, including: Strategies to reduce the flammability of organic phase change Materials: A review;  Triazine derivatives as organic phase change materials with inherently low flammability;  Bio-based materials for fire-retardant application in construction products: a review. This will help the readers to understand the recent progress of the efforts to reduce the flammability of material in buildings.  

4.     Some of the reference format need to be corrected, such as “2022” in reference 5, bold year, like “2013” in reference 17, 18, 19. Also checking “n.d.” in reference 32

Author Response

Dear Reviewer 3,

Thank you for your feedback. We have made your recommended edits and believe they have improved the manuscript. 

Round 2

Reviewer 2 Report

The authors corrected some mistakes identified in the first review round. However, many issues still require appropriate correction, as follows:

1. The authors provided incorrect numbering of subsections (e.g.; section 2.5 Nanoclays; section 2.6 Coatings, Paint and Drywall; section 3.5 Coatings and Composites; etc).

2. It seems that the word “weldability” was not removed from Section 3.1 (“Concrete”). What is the relationship between concrete and the weldability property cited in line 121? Is this property associated with the content of this subsection?

3. The subsections “Coatings, Paint and Drywall” and “Coatings and Composites” seem to cover similar contents. The organization of the content of these subsections should be improved, so that different sections will present distinct information.

4. The content provided in line 124 of the paper (“A novel and promising development in nanotechnology in concrete their ability to impart self-sensing properties”) should be supported with citation of relevant references. The references cited at the end of this paragraph (references 14 and 15) discuss the effects of post-fire-curing on plain and nanomodified cementitious materials. However, these studies did not address self-sensing properties of fire-damaged concrete. This paragraph should be complemented with a review of the effects of nanomaterials on the self-sensing properties of smart cementitious materials subjected to fire condition (e.g., Dong et al. 2020 - 10.1016/j.cemconcomp.2020.103675; Nalon et al. 2021 - DOI: 10.1016/j.cemconcomp.2021.104104; Jang et al. 2022 - 10.1016/j.jobe.2021.103816), which is a novel and promising application of nanotechnology on structural fire safety.

5. In the response to comment 10 of the first review round, the authors stated that the paper was enhanced with additional literature on the hazards associated with specific nanomaterial used in construction. However, they did not present relevant changes in the section “Hazards Associated with Nanomaterials Used in Construction”. This section should be significantly improved because it did not revise hazards associated with nanomaterials from the building construction perspective. It only presented a list of health issues associated with different types of nanomaterials. Health AND ecological hazards associated with nanomaterials must be revised in terms of potential damages during the different steps of the construction phase and during the useful life of building environments.

6. The authors response indicated that Figure 1 was removed. However, the revised version of the paper still presents this figure. Figure 1 is very simple and should be removed. All information provided in this image can be clearly described in the text of the paper.

7. English writing must be carefully revised. There are many grammar, typo and punctuation mistakes in the manuscript such as “a fire., as”; “While the application of (…) have been reviewed”; “nanomodified material Fire safety glass”; “material maymay be harmful”; “available in the literaureliterature”; “and & economic”; “impactsustainable material”; “SafetySustainability: Safety”; “However, Aalthough”.

Author Response

Dear Reviewer Two,

Thank you for your feedback. It appears most of the recommendations are related to trouble viewing track changes in MS Word. However, we believe we addressed your other comments and believe the manuscript has been sufficiently improved. Attached are responses to feedback.
